# Chilean Disaster Response and Alternative Measures for Improvement

**Luciana das Dores de Jesus Da Silva** [1,*], **Susanne Kubisch** [2], **Mauricio Aguayo** [3], **Francisco Castro** [4], **Octavio Rojas** [3], **Octavio Lagos** [5] and **Ricardo Figueroa** [1,*]

1 Department of Aquatic Systems, University of Concepción, Concepción 4070409, Chile
2 Department of Geography, University of Innsbruck, 6020 Innsbruck, Austria; susanne.kubisch@uibk.ac.at
3 Department of Urban Planning, University of Concepción, Concepción 4070409, Chile; maaguayo@udec.cl (M.A.); ocrojas@udec.cl (O.R.)
4 Earth Science Departament, Chemical Sciences Faculty, University of Concepción, Concepción 4070409, Chile; franccastro@udec.cl
5 Water Resources Department, University of Concepción, Concepción 4070409, Chile; octaviolagos@udec.cl
* Correspondence: lucisilva@udec.cl (L.d.D.d.J.D.S.); rfiguero@udec.cl (R.F.)

**Abstract:** Effective DRM aims to identify and minimize both hazards and vulnerabilities of a territory. This case study carried out in Chile analyzes national programs and disaster risk management structures at different administrative levels (national, regional, and municipal) and identifies gaps that contribute to the vulnerability of the current system. The proposed measures and options for improvement presented in this study are based on a literature review of scientific discussions about international governance, disaster risk management, and case studies conducted in Chile. The results indicate that the national disaster risk management plan has been adjusted in recent years, especially after the 2010 Chilean earthquake. The national administration, which is primarily responsible for managing potential risks, as well as the regional and local governments, has been replaced by the National Disaster Prevention and Response System (SINAPRED) in 2021, according to the 21364 law. This law was created to make cities more resilient, contributing to the Sustainable Development Goals (SDGs). This change is intended to decentralize disaster risk management, considering local conditions and preventing oversight of disaster risk management, which is not mandatory at the local level. It has also noted some gaps, such as the lack of standardization of emergency and early warning systems and funding at local levels. It is hoped that the system will move forward in this transition period and that the gaps will not affect effective risk management, as they have caused loss of life in past disasters.

**Keywords:** disaster risk management; Chile; early warning; local level; SDGs planning; Law No. 21364

## 1. Introduction

In light of the increase in natural disasters worldwide and the growing uncertainty and complexity of challenges that today's society has to face (Wassenius and Crona 2022; Jeworrek 2018), effective disaster risk management (DRM) is of utmost importance to save lives. Building a more resilient society and preparing the world population for present and future environmental threats have been concerns of scientists around the world for many years (EIRD 2004; United Nations Office for Disaster Risk Reduction 2015; Blaikie et al. 1994). While research on DRM cycles and recommendations for effective DRM plans are important, it is necessary to investigate the particularities of each system and question the dominant status quo of top-down Disaster Risk Reduction (DRR) in order to effectively deal with natural disasters, as local conditions differ between communities (Herrmann Lunecke 2015; Le De et al. 2013; Wisner et al. 2015).

Centralized management policies/systems utilize single communication channels, and disasters may disrupt those channels. Therefore, decentralized policies do not address

this issue due to the allocation of responsibilities (Amaratunga et al. 2019). Delegating responsibilities among different administrative levels can present a challenge, but it can be the key to an efficient regulatory system (Amaratunga et al. 2019; United Nations Office for Disaster Risk Reduction 2015).

In Brazil, for example, disaster risk management (DRM) is carried out by the National Civil Defense System (SINPEDEC), which has undergone adjustments influenced by the United Nations' International Conference on Environment and Development (Rio 92). It was further updated in 2005, the same year as the Hyogo Conference on Risk Reduction. However, it was not until 2010 that the National Civil Defense Policy was established as national law following the major landslides that claimed the lives of over 200 people in the State of Rio de Janeiro in 2009. Two years later, in 2011, another major landslide occurred in the same region, considered the worst disaster in Brazil, resulting in the deaths of over 1500 people, which raised questions about the recently enacted law. Following this disaster, the law was revisited, and numerous adjustments were made until it reached its current form. Despite the improvements brought about by these changes, the system remains centralized, considering the size and scale of the country (Ribeiro 2017).

Mexico also underwent a series of public measures regarding disaster risk management following an earthquake that struck the country in 1985, causing widespread destruction in Mexico City. Over the years, the National Civil Protection System (SINAPROC) has incorporated new elements and institutions, becoming one of the first systems to establish financial mechanisms for disaster prevention and response. Despite these improvements, it is recognized that there is a significant disparity between theory and practice when it comes to resilience issues. Bureaucracy has proven to be a major obstacle, and in terms of regulation, there is an evident gap between agreed-upon measures and their implementation. Integrated Risk Management was only incorporated into the regulations in 2012, seven years after its international recognition.

Some acknowledged weaknesses of DRM in Mexico include an inadequate early warning system and centralized management without proper distribution at different administrative levels (Wagle 2020; Cen 2017). Decentralizing disaster risk management (DRM), focusing on the local level, and establishing an efficient regulatory system are challenges for developing countries (Amaratunga et al. 2019; United Nations Office for Disaster Risk Reduction 2015; Uddin et al. 2021).

Located in the Pacific Ring of Fire, Chile has experienced two of the largest seismic events in history: 21–22 May 1960 (magnitude Mw = 9.5) and 27 February 2010 (magnitude Mw = 9.5) (Cifuentes 1989; Miranda et al. 2012). These earthquakes highlighted two conditions: weaknesses in the disaster risk management system and responsiveness and the need for changes in the administrative structure (Lagos and Gutiérrez 2005; Vilches et al. 2014). Other events, such as floods (Cubelos et al. 2019), landslides (Korup et al. 2019), fires (Bowman et al. 2019), extreme precipitation (Meseguer-Ruiz et al. 2020), and megadrought (Muñoz et al. 2020) are also part of the dynamics of the Chilean territory. Man-made disaster risks (Alaniz et al. 2019), social disasters (Louw et al. 2022), or industrial accidents generated by natural disasters (NATECH) (Molinos-Senante et al. 2023) also improve significant risk on the territory.

The National Emergency Office of the Ministry of the Interior, known as ONEMI, is responsible for risk management at the national level (Ministerio del Interior 2002). ONEMI has multiple offices distributed throughout the country. They receive information from various monitoring systems (Mizutori 2020) and, in the event of a hazard event, issue alerts (Ministerio del Interior 2002). The management of this was established by the National Civil Protection Plan, published in 2002 (https://www.bcn.cl/leychile/navegar?idNorma=199115, accessed on 20 August 2023), and the only law related to natural disasters was the Catastrophes Law published in 1965 (Ley 16828—https://www.bcn.cl/leychile/navegar?idNorma=214428, accessed on 17 July 2023) which focuses mainly on seismic hazards.

The previous territorial approach of Chilean DRM, represented by ONEMI, was insufficient, as it did not consider the local/regional hazards and vulnerabilities investigation in

Chilean territory (Riveros et al. 2023). The DRM presents problems such as the centralization of decision-making power and the lack of continuity between administrative levels (national, regional, and local) (Riveros et al. 2023). The lack of standardization of early warning systems has cost lives (Contreras Gatica and Sickinger 2016), as well as the absence of a single and direct legal instrument that establishes clear DRM guidelines and considers the geographic approach of the territory (Maskrey et al. 1993; Fuentealba and Verrest 2020).

Chile has created a new law that establishes the National Disaster Prevention and Response System, replaces the National Emergency Office with the National Disaster Prevention and Response Service, and adapts norms as indicated (Ley 21 364—https://www.bcn.cl/leychile/navegar?idNorma=1163423, accessed on 20 July 2023) with the objective of correcting the problems related to the previous system, proposed by ONEMI. This law is part of Chile's commitment to the Sustainable Development Goals (SDGs) to build resilient cities, reducing vulnerabilities and hazards (natural, biological, or anthropogenic) impacts in the territory (https://sustainabledevelopment.un.org/content/documents/15 134Chile(spanish).pdf, accessed 25 November 2023).

The new legal framework establishes the National System for Disaster Prevention and Response (SINAPRED), comprising a collection of public and private entities with competencies related to the different phases of the disaster risk cycle, and the National Disaster Prevention and Response Service (SENAPRED). These entities are organized in a decentralized and hierarchical manner, spanning the communal, provincial, regional, and national levels. The objective of this organizational structure is to offer effective disaster risk management, encompassing regulations, policies, plans, and other instruments and procedures related to disaster risk management (Ley 21364—https://www.bcn.cl/leychile/navegar?idNorma=1163423, accessed on 20 July 2023).

The new service, which assumes the responsibilities and rights of ONEMI, will supervise all emergency or disaster stages. It will be composed of public and private entities and will be guided by regulations, policies, plans, resources, and management instruments organized from the local to the national level, with a strong focus on prevention. The objective of this new system will be to promote and implement actions for the prevention, response, and management of emergencies that result in or have the potential to cause collective harm to people, property, or the environment.

This study reflects on Chile's progress in risk and disaster management policy, analyzing the Chilean DRM structure presented by the National Emergency Officer (ONEMI), identifying gaps in the process, and addressing issues related to incorrect early warnings in previous events before the new law. Scientific papers, urban planning instruments, normative files, and international reports were reviewed and analyzed for this purpose. Based on the results of the analysis, the weaknesses and strengths of the new system have been examined, particularly at the local level within the Chilean DRM framework.

This research aims to show the DRM transition in Chile and contribute to other communities in the world, reducing vulnerabilities and hazard impacts and improving resilience in the territories. It was concluded that Chile's public policies are in a transitional stage and internal readjustment stage of the risk system. Although internal changes are happening in governance form, it is also recommended to include interdisciplinary multi-hazard focus, sustainability, and local knowledge.

## 2. Methodology

Currently, DRM in Chile is undergoing a transition from a centralized system to a system that seeks to include the administrative levels (national, regional, provincial, and communal). In this new stage, the aim is to consider the local particularities of the diverse Chilean territory. The elements of risk should be considered, such as hazards (natural, biological, or anthropogenic), vulnerabilities, and exposure. However, before evaluating this transition, it is necessary to understand how a DRM is performed effectively (Section 2). Then, it describes how Chile's risk management system worked before the law, highlighting weaknesses in the over-centralization of the DRM, early warning, and response system.

New perspectives are recounted with the birth of the new disaster risk law for Chile. Finally, the gaps in the new law are discussed (Section 3). From a collection of approximately 50 documents, including articles, urban planning instruments (Metropolitan Plans), normative files (("Seismic and catastrophe cases—Law n° 16282 (1965)"; "Institutional Organization of States of exception—Law n° 18415 (1985)"; DL n° 369 (1974)—Organic Law ONEMI; DS n° 156 (2002)—National Plan of Civil Protection; etc.) and international reports. Sendai Framework Disaster risk governance measures recognize that the State has the primary role in reducing disaster risk but that responsibility should be shared with other stakeholders, including local government, the private sector, and other stakeholders) (https://www.undrr.org/implementing-sendai-framework/what-sendai-framework, accessed on 20 July 2023); United Nations Office for Disaster Risk Management *convenes, partners, and coordinates activities to create safer, more resilient communities* (https://www.undrr.org/terminology/disaster-risk-management, accessed on 20 July 2023); Accountability in the Context of Disaster Risk Governance details the meaning of responsibility in governance: A series of measures were found that suggest effective disaster risk management (DRM).

*How Effective DRM Is Proposed*

The Sendai Framework outlines four priorities for effective disaster risk management (DRM): (1) Understanding disaster risk, (2) strengthening disaster risk governance to manage disaster risk, (3) investing in disaster risk reduction for resilience, and (4) increasing disaster preparedness for effective response and "building back better" in recovery, rehabilitation, and reconstruction efforts. The first step is to comprehend the issue and subsequently establish governance measures (Bizikova et al. 2019; United Nations Office for Disaster Risk Reduction 2015).

Disaster risk governance measures should be designed to encompass national, regional, and local levels. The local level plays a crucial role in ensuring effective and efficient disaster risk management (DRM) (Ungera et al. 2020; Di Ludovico and Lodovico 2020). This entails establishing clear objectives, plans, competencies, guidelines, and coordination within and across sectors (applied in Malaysia (Hassan et al. 2020)) while also involving relevant stakeholders (applied in Sri Lanka (Saja et al. 2021)). Therefore, it is essential to strengthen disaster risk governance for prevention, mitigation, preparedness, response, recovery, and rehabilitation. Additionally, it is necessary to promote collaboration, partnerships (applied in Egypt (Badawi and Abdullah 2022)), and the active participation of local communities (applied in Malaysia (Yusoff and Yusoff 2022)) and institutions in implementing relevant instruments for disaster risk reduction and sustainable development (Ali et al. 2021; Camus et al. 2016; United Nations Office for Disaster Risk Reduction 2015; Journal Officiel de la Republique Algerienne n° 84 2014; Blaikie et al. 1994).

Another concept emphasized by the United Nations Office for Disaster Risk Management (DRM) is accountability within the context of disaster risk governance. In other words, those entrusted with the responsibility of managing various aspects of governance must be held accountable for fulfilling their obligations to ensure meaningful DRM. Accountability typically implies a legal obligation on the part of individuals in political, bureaucratic, or technical positions to carry out specific, clearly defined responsibilities or functions. If these individuals neglect their obligations and fail to perform their expected functions without justifiable reasons, they can be held accountable for their actions or omissions. To establish effective DRM, several measures were proposed to governments, following Amaratunga et al. (2019):

- Mainstreaming DRR into overall national policies: Decentralizing DRM from a single ministry or institution and transferring responsibility to other ministries through integrated work. The development and other relevant sectoral policies of the State need to contribute to reducing disaster risk. Enabling legislation: A review of existing legislation in relevant areas may be necessary to identify any legal obstacles to DRR or to create new legislation;

- Institutional development: Efficient institutional arrangements are essential for the creation, implementation, and revision of a plan. It is necessary to identify gaps, deficiencies, weaknesses, and institutional requirements, including technical capabilities. With the enhanced regulatory and coordination role of the State in the new DRR framework, developing the necessary institutional infrastructure is a critical step in ensuring that governments and other accountable entities meet public expectations regarding DRR (Chatiza 2019);
- Adequate resources: Strategically distributing and allocating resources is crucial and requires careful planning. In countries with limited or no resources, it is important to establish priorities and avoid duplicating functions among institutions. This can serve as an alternative to overcome some of the resource constraints. Empowerment of stakeholders: Establishing lines of authority and clarifying the different responsibilities of different sectors and levels of government can ensure timely actions. Emphasizing the importance of academic contributions through multidisciplinary research and involving stakeholders in consultations at various levels, including government, non-governmental organizations, academics, communities, and local officials, is essential;
- Regular monitoring, evaluation, and review: Given the diversity of actors and actions involved in DRR, such as political leaders, ministries, national institutions, sub-national and local authorities, the private sector, professional groups, and civil society organizations, regular monitoring, evaluation, and review of DRR processes and outcomes are of critical importance (Mizutori 2020);
- Community-based disaster risk management: People living in areas experiencing frequent natural hazards are not helpless victims but rather innovative agents who possess valuable Local Knowledge (LK). Local people have significant knowledge about hazards that affect their daily lives and their vulnerabilities, and they utilize this Local Knowledge to prepare for, mitigate, respond to, and recover from the impacts of disasters (Pixley et al. 2022; Šakíc Trogrlić et al. 2021; Di Ludovico and Lodovico 2020).

The local community extends beyond the group of people living in each area and includes indigenous populations. Over time, many traditional aspects of indigenous life have been suppressed and progressively extinguished. These aspects encompass beliefs, knowledge, and practices that contribute to disaster risk reduction (DRR) capacities, including those that enable coexistence with natural processes due to their long history of living in harmonious and interconnected relationships with nature (Ali et al. 2021).

Assigning responsibilities to a national-level administration is necessary, and capabilities are crucial. It requires a clear delineation of roles and responsibilities regarding the generation and transmission of information in an effective and timely manner. The incorporation of Local Knowledge (LK) is cost-effective as it leverages local capacities and can reduce the reliance on external assistance (Amaratunga et al. 2020; Šakíc Trogrlić et al. 2021).

This section describes how a DRR works effectively based primarily on international guidelines and scientific publications. The following sections will present the context of Chile in this process based on national documents.

## 3. Results and Discussions

This section explains the Chilean context, considering disaster risk management (DRM) before the implementation of the new legislation. Subsequently, the new law is presented, highlighting the main changes it brings, especially in the context of urban planning. Some limitations are highlighted, from low: (1) Standardization of the emergency between the monitoring institutions and the host institution; (2) accountability is an integral part of good governance. Those entrusted with the responsibility of managing various aspects of governance should be held accountable for what they are expected to perform to ensure meaningful disaster risk reduction; (3) the new legal instrument does not make clear what the sources of funding are and how resources should be distributed to achieve the objectives at different levels, and (4) how decentralization will succeed.

*3.1. How DRM Works in Chilean Territory before New Legislation and the Distribution of Risk and Vulnerability*

Regulatory tools play a fundamental role in disaster risk management (DRM). In Chile, DRM was primarily based on the National Civil Protection Plan, published in 2002, which establishes a management structure. The National Emergency Office of the Ministry of the Interior, known as ONEMI, was responsible for risk management at the national level (Ministerio del Interior 2002). ONEMI had multiple offices distributed throughout the country. They received information from various monitoring systems and, in the event of a hazard event, issued alerts. ONEMI was also responsible for planning, coordinating, and executing actions aimed at preventing or addressing problems arising from earthquakes or disasters (Plan Nacional de Emergencia 2017; Herrmann Lunecke 2015).

The previous system was organized as follows: The country had an early warning system that included a network of sensors and monitoring to prevent or mitigate the effects of natural disasters, such as earthquakes and tsunamis (focus) (Plan Nacional de Protection Civil 2002). The Civil Protection Plan promoted education and training of the population in disaster prevention and response. In the case of a disaster, Chilean authorities coordinated the response through the National Civil Protection System (Ministerio del Interior 2002). This system involved different state agencies, such as the National Emergency Office (ONEMI), the Ministry of Health, and the Armed Forces, among others, to ensure an effective response (Decreto de Ley 369 (1974)—https://www.bcn.cl/leychile/navegar?idNorma=6027, accessed on 17 May 2023). Chilean public policy efforts on the identification and reduction of risks through territorial planning and the regulation of buildings to withstand earthquakes, and in the event of a disaster, an Emergency Operations Committee (COE) was activated to coordinate response and later recovery actions. The affected population received support through public resources and services (Plan Nacional de Protection Civil 2002, https://www.bcn.cl/leychile/navegar?idNorma=199115, accessed on 17 May 2023).

Although ONEMI's main objective was to maintain a decentralized approach to disaster risk management (DRM) and to serve as a structured basis for regional, provincial, and communal planning (Ministerio del Interior 2002), decisions are mostly concentrated at the national level. The regional level was responsible for coordination and mediation between the national and local levels. Regional administrations were in charge of assisting local administrations with disaster mitigation implementation, preparedness, response, recovery, and livelihood development programs and projects, which could be planned and supervised by national institutions (Ministerio del Interior 2002).

The administrative sphere presents issues such as the centralization of decision-making power and a lack of continuity between administrative levels (national, regional, and local). There is also a lack of standardization of early warning systems and the absence of a single, direct legal instrument that establishes clear DRM guidelines (Maskrey et al. 1993). Risk analysis and assessment appear as part of risk management in the Plan Nacional de Protection Civil (2002). Unfortunately, it is not clear whether these stages should contemplate the three administrative levels. Theoretically, the concepts of prevention, preparedness, and response are also present in the plan under the same conditions as the latter (Ministerio del Interior 2002).

According to the Plan, in the case of prevention, the aim is to suppress or definitively prevent natural or human-generated events from causing damage through the eradication of dwellings from at-risk locations. In practice, there are challenges to the implementation of these measures since local regulations allow the occupation of areas (Camus et al. 2016; Valdivieso 2012). A set of measures and actions, such as an Inventory of human and financial resources, the preparation of Response Plans, determination of coordination and its procedures, drills, and simulations, the training of personnel and the community, operational training, and providing information to the community comprise the pre-law preparedness actions. Basically, it was mostly perceptive that ONEMI's actions were

focused on the response to an event, which also presented problems (Merten and Jaque 2022).

The 2010 earthquake highlighted failures in the transmission of tsunami early warning system information to certain locations (Vilches et al. 2014), even though these aspects were described in the plan. The inability to provide a desirable response following the event prompted a reformulation of DRM in the Chilean Congress (Camus et al. 2016; Vilches et al. 2014). However, this reformulation has been insufficient, and it is crucial to consider holistic aspects of the administrative structure (Bresciani Lecannelier 2012).

Chile's location on the Pacific Ring of Fire has caused the country to experience two of the largest seismic events in history: 21–22 May 1960 (magnitude Mw = 9.5) and 27 February 2010 (magnitude Mw = 9.5) (Cifuentes 1989; Miranda et al. 2012). These earthquakes highlighted two conditions: deficiencies in the disaster risk management system and response capacity and the need for changes in the administrative structure (Lagos and Gutiérrez 2005; Vilches et al. 2014). Other events, such as floods (Cubelos et al. 2019), landslides (Korup et al. 2019), fires (Bowman et al. 2019), extreme precipitation (Meseguer-Ruiz et al. 2020), and megadrought (Muñoz et al. 2020), are also part of the dynamics of the Chilean territory. Man-made hazards (Alaniz et al. 2019), social disasters (Louw et al. 2022), or industrial accidents generated by natural disasters (NATECH) (Molinos-Senante et al. 2023) also increase the risk on the territory.

According to the Intergovernmental Panel on Climate Change (IPCC), Chile is one of the most vulnerable countries in the world to the impacts of climate change. Situated in South America, Chile possesses diverse natural ecosystems (deserts, mountain ranges, glaciers) that are affected by its economic activities. Chile has made commitments to modify its actions and mitigate the effects of climate change on its territory. The United Nations Framework Convention on Climate Change (UNFCCC) has established nine criteria to classify a country as vulnerable, and Chile meets seven of them (Ministerio del Medio Ambiente 2016; IPCC 2023). The criteria met by Chile include low-altitude coastal areas (Soto-Muñoz and Guerrero-Valdebenito 2022), arid and semi-arid areas, areas with forest cover and areas exposed to forest deterioration, areas prone to natural disasters, areas exposed to drought and desertification, areas of high urban air pollution, and areas of fragile ecosystems, including mountain ecosystems (Proyecto de Ley del Cambio Climatico de Chile 2021, https://www.bcn.cl/leychile/navegar?idNorma=1177286, accessed on 14 November 2023).

Vulnerability analyses, in general, are limited as most research tends to focus on quantifying the hazard. However, for risk management measures to be effective in a particular territory and to mitigate the impacts of catastrophic events, it is crucial to assess the vulnerabilities present in the area (Le De et al. 2015). The guidelines for describing, representing, or quantifying vulnerability will depend on the element exposed, the type of hazard, and the scale of the analysis (Corominas et al. 2014).

Hazards and vulnerability establish thresholds between security and risk in continental Chile (Vilches et al. 2014). The ability to control nature is limited, except in the realm of disaster prediction. Therefore, the only way to decrease the likelihood of disasters is by addressing vulnerability, as humans do not have equal access to resources and opportunities, nor are they equally exposed to hazards (Blaikie et al. 1994; Le De et al. 2015). Society's vulnerability can manifest through various components or elements because of specific social processes. Vulnerability is reflected in the locations where people reside, the production and infrastructure in areas prone to damage, the insecurity of structural buildings, as well as the lack of economic resources. To comprehend disasters, it is essential to not only understand the types of hazards that may impact people but also recognize the varying levels of vulnerability among different groups (Lavell 2001; Blaikie et al. 1994).

The centralized administrative structure presents an administrative challenge, as it fails to adequately consider the demands, needs, and unique local characteristics (social, ecological, economic) of its residents. Additionally, there are significant disparities in

monetary resources and capacities among different communes, resulting in difficulties in addressing the challenges faced by these communities (Brain and Mora 2012).

*3.2. Decision-Making Process*

In order to understand how the decision-making system works in case of an event, the structure of this process is detailed. At a local level, Emergency Operations Centers (COEs) are established at the national, municipal, and local levels. Their primary objective is to address potential emergencies, ongoing emergencies, or emergencies already occurring. The purpose is to establish and coordinate mitigation actions while providing support to regional decision-making. The technical tables follow the process outlined in Figure 1 (ONEMI—Ministerio del Interior y Seguridad Pública 2017):

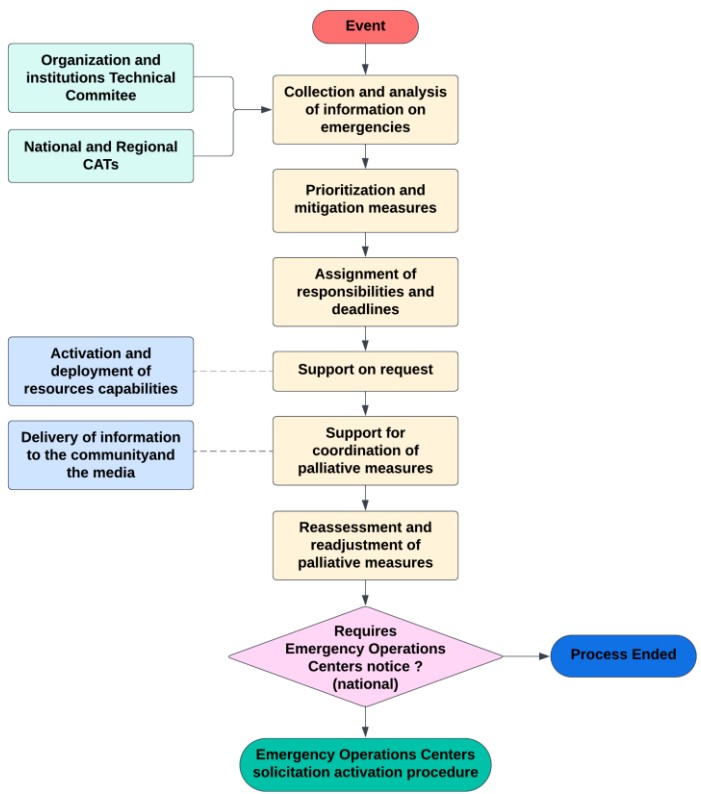

**Figure 1.** Early warning decision taking. Source: Reprinted/adapted with permission from Ref. [Plan Nacional de Emergencia (2017)]. Copyright 2023, copyright owner's Ministerio del Interior y Seguridad Pública (https://www.bcn.cl/leychile/navegar?idNorma=1106167, accessed on 22 December 2023).

In case of an event, ONEMI presented the following structure for decision-making and early warning issuance:

- Information gathering and analysis: This stage primarily involves collecting information from different territorial levels (communal, provincial, and regional), consolidated by the National Early Warning Centre (CAT) through standardized instruments of the National Civil Protection System (SNPC). It also includes information provided by organizations and institutions belonging to the technical roundtable. The analysis focuses on evaluating the emergency, its impact, and the damages caused;
- Prioritization of mitigation actions: Based on the gathered information and analysis of the emergency, mitigation actions are prioritized. Responsibilities and deadlines are assigned for each action;

- Support for requirements and mitigation: This stage involves providing support for regional requirements, activating and deploying resources and capabilities if necessary, and coordinating mitigation actions;
- Re-evaluation and readjustment of mitigation actions: This stage encompasses all the previous stages and includes the ongoing evaluation and adjustment of mitigation actions. It aims to improve the overall process and make necessary corrections;
- Delivery of information to the community and the media: This stage focuses on delivering information to the community and the media. Its objective is to reduce uncertainty among the population by providing updates on mitigation actions and coordinating efforts between the Technical Committee and the affected regional and/or provincial levels.

In the structure described above, community participation in decision-making was not considered in the decision tables. The entire process was concentrated solely within the technical department, with information then being disseminated to the community. However, risk reduction measures are most effective when they involve the direct participation of individuals who are most likely to be exposed to hazards. This includes their involvement in planning, decision-making, and operational activities at all levels of responsibility (Anderson et al. 2014).

*3.3. Operational Phase*

Prior to the law, ONEMI's operating system relied on a redundant information system. The CATs located in each region received information from five monitoring sources and six technical agencies: (1) SERNAGEOMIN: The National Service of Geology and Mining, responsible for generating, maintaining, and disseminating information on basic geology, geological resources, and hazards (SERNAGEOMIN 2023, Available in https://www.sernageomin.cl/mision-y-vision-institucional/, accessed on 20 November 2023). (2) CONAF: The National Forestry Corporation (CONAF) is tasked with managing Chile's forestry policy and promoting sector development (CONAF 2023, Available in https://www.conaf.cl/quienes-somos/mision-y-objetivos/, accessed on 20 November 2023). (3) SHOA: The Hydrographic and Oceanographic Service of the Chilean Navy provides technical elements, information, and assistance for navigational safety, including early warning for tsunamis (SHOA 2023, Available in https://www.directemar.cl/directemar/organizacion/mision-y-vision, accessed on 20 November 2023). (4) CSN: The National Seismological Centre (CSN) is the official technical agency responsible for monitoring seismic activity across the country (CSN 2023, Available in https://www.sismologia.cl/accesos/nosotros.html, accessed on 20 November 2023). (5) DGA/DOH: The state agency responsible for managing and administering water resources in a sustainable and efficient manner, providing information through its hydrometric network and Public Water Cadaster (DGA 2023, Available in https://dga.mop.gob.cl/acercadeladga/mision/Paginas/default.aspx, accessed on 20 November 2023). (6) DMC: Provides meteorological information and services to support various socioeconomic activities and conducts meteorological research in collaboration with national and international agencies, managing the National Weather Data Bank (DMC 2023, Available in https://www.meteochile.gob.cl/PortalDMC-web/index.xhtml, accessed on 20 November 2023).

These agencies collectively provided information to ONEMI for disaster management and response. In theory, the system operates through the following steps: First, the information collected by the institutions is sent to the SNPC, where it is classified into three different levels: green—preventive early warning, yellow, and red. Second, the SNPC determines whether measures such as evacuation are necessary. If the response is affirmative, a meeting is convened with the various administrative levels. Finally, the level of impact is determined based on the following categories: (1) Presumed catastrophe, (2) Rescue equipment, (3) Medical evaluation, (4) Isolated evacuation, (5) Critical infrastructure, and (6) Order and security (ONEMI—Ministerio del Interior y Seguridad Pública 2017).

Risk management at the local level is carried out through the Emergency Operations Centers (Centro de Operaciones de Emergencia—COE), which are physical facilities where the National Emergency Operations Committee operates. The COEs are activated in the event of an emergency to coordinate and provide a response to the situation (ONEMI—Ministerio del Interior y Seguridad Pública 2017). These Emergency Offices acted autonomously and had no links with the municipal public administration. In case of a hazardous event, public administration was notified (Ministerio del Interior 2002).

In practice, the early warning system exhibits inconsistencies in the transmission of information. There is a lack of standardized hazard classification between the various monitoring institutions and ONEMI. In certain cases, warnings are issued for areas without actual danger or are delivered after a hazardous event, as was the case with the tsunami event in 2010 or after the Maule earthquake (Herrmann Lunecke 2015). The failure of ONEMI's early warning response in communities such as Arauco and Tubul (Biobío region) has highlighted the need for continuous reinforcement of early warning systems for hazardous natural events. This demonstrates a weak emergency structure and deficient coordination among different institutions (ministerial, public, and private) that have separate roles (Arenas Vásquez et al. 2012; Vilches et al. 2014).

*3.4. Legal Background, Urban Planning, and New Perspectives*

The legal instruments followed by ONEMI for risk management in Chile do not specify premises, objectives, concepts, or assessment structures or provide clear attributions and responsibilities (Decreto de Ley 369 (1974)—https://www.bcn.cl/leychile/navegar?idNorma=6027, accessed on 20 July 2023). The National Plan of Civil Protection states that "ONEMI was created as a centralized public service with the mission of planning, coordinating, and executing actions aimed at preventing or addressing problems arising from earthquakes or disasters" (Riveros et al. 2023). The first specific objective states that this national civil protection management framework should be based on a decentralized administration perspective, serving as a structured foundation for regional, provincial, and communal planning, taking into account the respective realities of risks and resources (Ministerio del Interior 2002). The centralization of public services poses a challenge to the decentralization of the administration of risk and disaster management systems across different levels (Riveros et al. 2023).

The National Disaster Prevention and Response System (SINAPRED), hereafter referred to as "the System", was published on 7 August 2021. The System consists of a collection of public and private entities with competencies related to the different phases of the disaster risk cycle. These entities are organized in a decentralized and staggered manner, spanning from the communal, provincial, regional, and national levels. The purpose of the System is to ensure adequate Disaster Risk Management, including the establishment of norms, policies, plans, and other instruments and procedures related to Disaster Risk Management.

SINAPRED was created to replace the National Emergency Office (ONEMI) in the National Disaster Prevention and Response Service. The new law defines six basic elements for disaster risk management: Hazard, emergency, emergency levels, disaster risk management, disaster risk reduction, and vulnerability. The system is based on three management plans:

1.  The National Strategic Plan for Disaster Risk Reduction;
2.  Disaster Risk Reduction Plans at the regional, provincial, and communal levels, which are implemented during the Mitigation and Preparedness Phases;
3.  Emergency Plans and their annexes, which are activated during the Response Phase at all levels.

These plans should be coordinated and prioritized at the national, regional, and communal levels to ensure effective systematization and comprehensive coverage (Ministerio del Interior y Seguridad Pública, Chile 2021). The law establishes that disaster risk management should follow a coordination structure (Ley 21364—https://www.bcn.cl/

leychile/navegar?idNorma=1163423, accessed on 20 July 2023). The disaster risk management committees are created to fulfill the functions of each phase at the national, provincial, regional, and communal levels, as appropriate. During the mitigation and preparedness phases, these committees will approve the disaster risk management instruments established by this law and will coordinate the necessary actions to enhance capacities and allocate resources for strengthening disaster risk management. The composition of the committees will be determined based on the level of emergency in the affected geographical areas in order to carry out the functions related to the response and recovery phases. The Ministry of the Interior and Public Safety will issue regulations that define the operation and frequency of these committees in each phase (Ley 21364—https://www.bcn.cl/leychile/navegar?idNorma=116342, accessed on 20 July 2023).

*3.5. Operational Phase*

The operational phase of DRM has undergone significant changes. Two technical agencies, the Chilean Fire Department and the Chilean Nuclear Energy Commission, will be included in the current monitoring system. Furthermore, the law states that, if necessary, agreements can be made with national or international organizations involved in threat monitoring (Figure 2). However, there is no information regarding the standardization of these new risk levels between the receiving and issuing agencies (Plan Nacional de Protection Civil 2002, https://www.bcn.cl/leychile/navegar?idNorma=199115, accessed on 20 July 2023; Ley 21364—https://www.bcn.cl/leychile/navegar?idNorma=1163423, accessed on 20 July 2023).

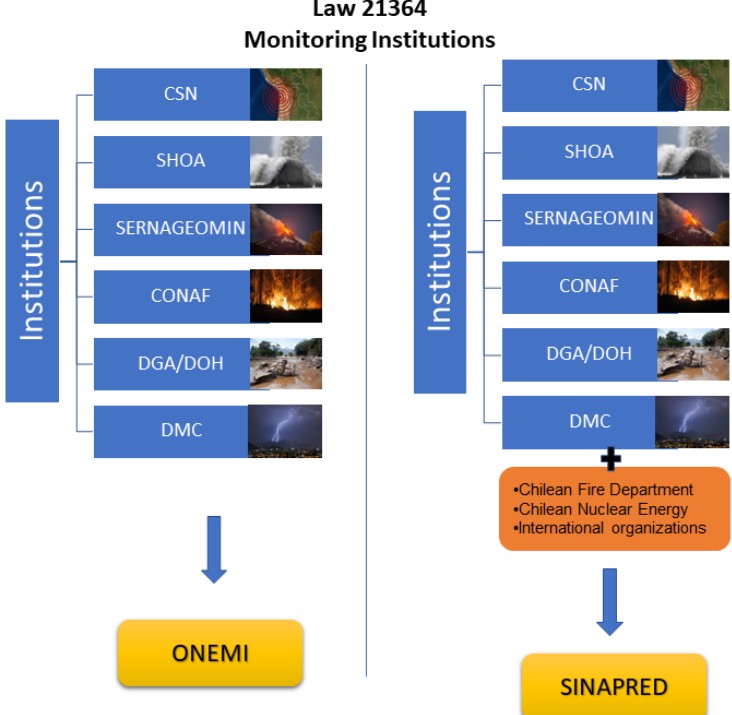

**Figure 2.** Previous and current early warning structure; SERNAGEOMIN: The National Service of Geology and Mining; CONAF: The National Forestry Corporation; SHOA: The Hydrographic and Oceanographic Service of the Chilean Navy; CSN: The National Seismological Centre; DGA/DOH: General Water Directorate/Directorate of Hydraulic Projects; DMC: Meteorological Directorate of Chile.

It is important to note that emergency levels are determined based on the evaluation of various factors and conditions, including the extent of the affected area, the number of people affected or potentially affected, and the response capacity of the administrative

levels involved. The levels are classified as follows (Ley 21364—https://www.bcn.cl/leychile/navegar?idNorma=1163423, accessed on 20 July 2023):

i.      Minor Emergency: Impact can be administrated by using communal capabilities and potentially with reinforcements or support from other areas through coordination at the communal level;

ii.     Major Emergency: Impact can be managed using regional capabilities and potentially with reinforcements or support from other areas through coordination at the provincial or regional level;

iii.    Disaster: Impact cannot be managed using regional capabilities alone and requires reinforcements or support from other areas of the country through coordination at the national level;

iv.     Catastrophe: Impact and devastation that exceeds the country's capacities and requires international assistance. Coordination at the national level is necessary to mobilize and coordinate international support.

### 3.6. Urban Planning and DRM

It is evident that the current urban planning instruments lack clearly defined DRM guidelines at the local administration level, and they are not addressing the first objective of the National Civil Protection Plan. The Municipal Plans (PLADECO—Plan de Desarrollo Comunal), which serve as indicative instruments for each municipality and should guide the development of the commune, do not establish a direct connection with the regional offices and do not impose management obligations at the municipal level (Valdivieso 2012).

The urban planning instruments did not include the creation of DRM departments at the municipal level, making their existence non-mandatory. As a result, many communes were left vulnerable to hazards, and there was a lack of proper monitoring. In several cases, early warnings were not issued on time after an event occurred, such as the significant landslide that took place in the city of Chaitén, located in the Chilean Patagonia, resulting in casualties and injuries (Morales et al. 2021).

The new law introduces significant changes for Chile in this regard. The first noteworthy point is that DRM is now incorporated at all levels (national, regional, provincial, and communal). The law mandates that DRM must be integrated into the management instruments at the municipal level, and each municipality must establish a department responsible for local risk management. This is a crucial provision that recognizes the need to consider the unique characteristics of each territory in Chile. It is also worth emphasizing the importance of citizen participation in local management processes, which is also addressed in the new law. This new DRM framework propels Chile towards a more comprehensive approach, considering the social contexts involved (Choi et al. 2019).

### 3.7. Talcahuano City—A Good Practice Example for Risk Management in Chile

Talcahuano has become the first commune in Chile to receive a national award for its innovative and inclusive approach to DRM. Located in the Biobío region's Concepción province, Talcahuano is a national leader in local risk monitoring and DRM. Given its geographical position on the Pacific Ocean, Talcahuano is susceptible to various hazards, including tsunamis, seismic events, river floods, landslides, and forest fires (Yévenes-Subiabre 2019; Bull 2022). Community leaders in Talcahuano have implemented strategic procedures for risk reduction, including (1) territory characterization, (2) identification of threats, (3) participatory diagnosis, (4) vulnerability analysis, (5) capacity analysis, and (6) risk scenario construction. This information is available through various media channels, including the city's website (http://www.talcahuano.cl/minisitio/gestion-de-riesgo/, accessed on 20 July 2023) and scientific articles that showcase the progress made in territorial risk management and risk mapping (Marshall 2020; Li et al. 2021).

The establishment of the Department of Disaster Risk Management in Talcahuano was driven by motivated and capable local municipal leaders in the aftermath of the 2010 earthquake and tsunami. The department focuses on educating the population and miti-

gating the impact of potential future disasters through drills, sirens, and community cell phone alerts (Herrmann Lunecke 2015). The local administration of Talcahuano has implemented preventive measures, disseminated information on fire, tsunami, and earthquake prevention in schools and neighborhoods, and created guides for volunteers in emergency situations (Soto-Muñoz and Guerrero-Valdebenito 2022). They have also published books and informative materials for the community. Prior to the new legislation, Talcahuano's local administration took the initiative to establish anti-tsunami construction requirements, which were not mandatory in other plans within the same region (Yévenes-Subiabre 2019; Saravia and Aguilar 2023).

The combination of capable leadership and political will mobilized by historical disasters is not common in Chile. This makes Talcahuano an outlier and a leader in the context prior to the new law.

### 3.8. Risk and Disaster Assessment Mandatory and Regulated by Law

The new legal instruments bring enforceability to DRM efforts. The creation of legislation that addresses specific issues is crucial to ensure the legitimacy of an effective DRM plan, as these actions should guarantee its incorporation into local planning instruments. The new plan's approach also emphasizes the decentralization of management across different administrative levels, similar to the previous plan. Therefore, one alternative is to establish a dedicated disaster risk management law in Chile. The structure of Chilean law encompasses the following aspects (Lei 12608 of Brazil 2012; Amaratunga et al. 2019; Ministerio del Interior y Seguridad Pública, Chile 2021). According to Amaratunga et al. (2019) and Liswanty and Prabowo (2021), six points should be highlighted:

1.  Concepts: Clear definitions of all risk elements such as hazard, vulnerability, susceptibility, exposure, risk, disaster, catastrophe, etc;
2.  Structure: Establishment of a comprehensive risk management framework with institutional bases and provisions, organized and detailed across different levels;
3.  Risk Assessment Process: Integration of the risk management process into organizational management, incorporating it into the institutional culture and practices and adapting it to internal processes;
4.  Surveillance: Implementation of control and supervision mechanisms to ensure that proposed measures are effectively carried out;
5.  Accountability: Definition of competencies and responsibilities for each institution involved in DRM, considering the different administrative levels of the country;
6.  Attributions and Responsibility: Identification of primary responsibilities for establishing, approving, maintaining, monitoring, and continuously improving the risk management policy, framework, and process. The relevant points of this law will be discussed in the following topics.

### 3.9. Improve Information Exchange between Monitoring and Reception

The institutions responsible for monitoring, information reception, and early warning should standardize emergency classification. This will allow for efficient transfer of information and help prevent inconsistencies in the early warning model. The decision tables that assess the post-event risk situation must consider the knowledge and participation of citizens. This process enables the community to express their concerns, exercise their rights and responsibilities, resolve conflicts, and collaborate in providing public services (Michels and Graaf 2010; Waheduzzaman and As-Saber 2015). In community-based disaster risk management (DRM) initiatives, addressing the vulnerability of communities to direct or indirect damage is often a primary focus in reducing overall risk. These initiatives may involve raising awareness about hazards, emergency planning, improving infrastructure reliability, and enhancing community resilience, including efforts to reduce poverty (Anderson et al. 2014).

*3.10. Top-Down Connection between the Three Different Main Administrative Levels (National, Regional, and Municipal)*

In general, hazard mitigation programs in developing countries are increasingly focusing on multilateral approaches for implementation (Anderson et al. 2014). The future structure aims to decentralize public administration to facilitate appropriate planning that addresses the specific needs of each level within public agencies. This includes enhancing community participation, defining the responsibilities of various actors, and standardizing risk assessment and the essential components of emergency plans (Aldunce and León 2007). At the local level of administration, there is potential for promoting a horizontal approach to DRM, fostering a collaborative relationship between civil protection agents and the vulnerable population, and encouraging community participation (Ribeiro 2017; Uddin et al. 2021). Community participation (Di Ludovico and Lodovico 2020), including Local knowledge (LK), is vital and draws upon the community's perception of hazards (Contreras Gatica and Sickinger 2016). Community leaders play a crucial role in taking action and disseminating their vision to other residents. (Anderson et al. 2014).

LK embodies participation, empowerment, and enhanced project sustainability (Dekens 2007; Šakíc Trogrlić et al. 2021). Acknowledging the value of LK is recognized as a means to improve the implementation of externally introduced interventions and technologies, especially in contexts where local governments and non-governmental organizations (NGOs) operate with limited budgets (Ali et al. 2021; Dube and Munsaka 2018; Howell 2003; Šakíc Trogrlić et al. 2021). From a process perspective, it encompasses how this knowledge is generated, acquired, refined, discussed, and perceived at the local level, thereby facilitating collective decision-making (Šakíc Trogrlić et al. 2021; Usón et al. 2016).

*3.11. Recommendation for National Decision-Makers*

Chile's DRM law provides an important framework for the country, addressing the main gaps from the previous situation. However, there are three important points to consider:

1. Standardization of emergency systems—Unfortunately, recent events have highlighted issues arising from the lack of standardization between monitoring institutions and the receiving institution (Vilches et al. 2014). Furthermore, the current law does not clearly specify whether the new classification is standardized across all institutions;
2. Accountability—The Sendai Framework recognizes that the State has the primary role in reducing disaster risk, but this responsibility should be shared with all stakeholders, including local governments and the private sector. Accountability is an integral part of good governance. Those entrusted with the responsibility for managing various aspects of governance should be held accountable for their expected actions to ensure a significant reduction in disaster risks (Liswanty and Prabowo 2021). Unfortunately, the law does not specify who will be held responsible in case of non-execution;
3. Funding—The new legal instrument does not clearly define the sources of financing or how resources should be allocated to achieve the objectives at different levels (Kousky 2019).

## 4. Conclusions

Chile, given its commitment to the SDGs, recognized the need to adjust its DRM system since the system prior to Law 21364 did not adequately consider socio-environmental vulnerabilities and limitations. Disaster risk management is a complex process that requires understanding various risk-related aspects, evaluating risks, and implementing appropriate management measures. It relies on accumulated knowledge from various fields and disciplines for evidence-based decision-making and policy development. Policies need to go beyond these aspects and consider the implications of vulnerability analysis in disaster risk reduction efforts.

The country has faced numerous natural disasters throughout its history, particularly the devastating earthquake in 2010, which revealed vulnerabilities and deficiencies in managing large-scale emergencies and providing essential services during such events. This article addresses the gaps in the previous DRM law in Chile and anticipates positive changes with the new law. Some important points from the new law include mandatory risk and disaster assessment, improved information exchange between monitoring and reception institutions, strengthened coordination between national, regional, and municipal levels, integration of local risk management into urban planning, community-based disaster management, and community participation.

The new law establishes three fundamental pillars for building a new National Emergency System: preventive work as the most effective way to save lives, an intersectoral approach and planning for effective solutions, and breaking down institutional silos to foster collaboration among stakeholders. The aim is to create a robust and modern institutional framework that reflects these principles: prevention, subsidiarity, and intersectionality.

This study emphasizes the importance of governmental measures to address territorial vulnerability caused by the lack of urban planning. These measures should be mandatory and tailored to local specificities. The local level extends beyond government administration, as it involves the strategies and local knowledge (LK) of communities, NGOs, and indigenous peoples in dealing with risks. LK is increasingly recognized in DRM, and their perspectives need to be carefully considered through dialogue between the state (at different levels), the private sector, local communities, and individuals. The impressions recorded in this study can provide experiences and new perspectives for change in countries that are going through a transition stage of risk management, as well as encourage the improvement of the disaster risk management system to build resilient cities as part of the SDGs, reducing the impact of global changes.

**Author Contributions:** Conceptualization, L.d.D.d.J.D.S. and S.K.; methodology, O.R.; software, L.d.D.d.J.D.S. and S.K.; formal analysis, L.d.D.d.J.D.S.; investigation, M.A.; resources, R.F.; data curation, L.d.D.d.J.D.S.; writing—original draft preparation L.d.D.d.J.D.S. and S.K.; writing—review and editing, L.d.D.d.J.D.S. and F.C.; visualization, O.L.; supervision, O.R.; project administration, R.F. and O.L.; funding acquisition, R.F. All authors have read and agreed to the published version of the manuscript.

**Funding:** This work was supported by the University of Concepción and CHRIAM (Centro de Recursos Hidrícos para la Agricultura y Minería)—ANID/FONDAP/15130015 and ANID/FONDAP/1523A0001.

**Institutional Review Board Statement:** Not applicable.

**Informed Consent Statement:** Not applicable.

**Data Availability Statement:** Data available in a publicly accessible repository that does not issue DOIs. Publicly available datasets were analyzed in this study. This data can be found here: [https://www.bcn.cl/leychile/navegar?idNorma=199115, accessed on 20 July 2023; https://www.bcn.cl/leychile/navegar?idNorma=1163423, accessed on 20 July 2023; https://www.bcn.cl/leychile/navegar?idNorma=1106167, accessed on 20 July 2023; https://www.bcn.cl/leychile/navegar?idNorma=214428, https://www.bcn.cl/leychile/navegar?idNorma=29824. Accessed on 20 July 2023].

**Acknowledgments:** We thankfully acknowledge the CHRIAM (Centro de Recursos Hidrícos para la Agricultura y Minería)—ANID/FONDAP/15130015 and ANID/FONDAP/1523A0001 providing funds to this investigation.

**Conflicts of Interest:** The authors declare no conflicts of interest.

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
