# Peer review of "Chilean Disaster Response and Alternative Measures for Improvement"

_socsci, doi:10.3390/socsci13020088_

Round 1

Reviewer 1 Report

Comments and Suggestions for Authors

Dear Authors,

In line with the proofreading criteria of the publisher, I prepared a reviewer’s report, which would be as follows:

The content of the proposed paper mostly meets the objectives of the journal.

Using the scientific methods (mainly literature review) for the preparation of the case study applied in accordance with the author’s scientific objectives resulted useful scientific achievements.

The main strength of the study is that the authors – in the form of case study and based on international experience - systemically introduce the national disaster risk management system including legal and institutional system.

The references used in the main chapters are relevant and assist the reader to understand the authors proposals. The illustrations used are regular and correct.

General comment is that, the article uses many abbreviations, which make some parts of the study difficult to understand. Therefore it is recommended to create a list of abbreviations for the article and annexed it.

In figure 2, it is definitely recommended to write the full names of the institutions instead of using the abbreviations.

In addition to acknowledging the high-quality work, I recommend the following minor corrections:

1. Introduction. At the end of Introduction, the main objectives of this study should be clearly and detailed presented, and the main conclusions should be highlighted.

The authors deal exclusively with events of natural origin (earthquakes, tsunamis), even though man-made disaster risks (chemical, transport or nuclear events), social disasters (Covid 19) or industrial accidents generated by natural disasters (NATECH) also prove to be a significant risk factors. The issue of flash floods is also a significant source of risk.

It is recommended to deal with man-made risk sources in addition to natural ones, or to delineate them in the introductory section.

2. Methodology. When describing the methodology, it is recommended to separate the international and domestic sources of analysis. In addition to subsection 2.1 of the article, a new subsection 2.2 should be included.

3. Results. 

It is recommended to present the  risk analysis procedure and risk reduction measures (based on national regulation) that form the basis of the risk management system in more detailed manner.

It is also recommended to explain the national, regional and local prevention, preparation, response and recovery measures applied for national disaster risk reduction in a more systematic way.

In addition to the organizational and management measures of risk reduction, it is also worth investigating and developing technical devices (for example early warning systems). An example of this is the following study (https://www.mdpi.com/2073-8994/13/8/1528 ). It is recommended for the authors to present some concrete national technical examples.

Sections 4., 5., and 6.

The “Discussion” part is missing from the article, where the authors examine their results in a broader scientific context.

It is recommended to define the relationship between section “3. Results” and sections 4-6 (including mainly discussion work) in an explanatory sentence in the introduction or methodology part of the article.

I suggest to rename Section "6.4. Important points" to "6.4. Recommendations for national decision-makers"

 7. Conclusions.

It is recommended to specify shortly the possibilities of international theoretical and practical applicability of the present case study research, as well as the possible directions of future research projects.

Based on the above, after minor revision, I suggest the publication of reviewed article.

Comments on the Quality of English Language

 Minor editing of English language required.

Author Response

We thank both reviewers for their time and constructive comments, which improve the quality of our manuscript significantly. The referee comments are formatted in “normal” and our responses are in bold and cursive with the mark “//”. In the manuscript the changes and additions are highlighted in yellow, green and blue colors, respectively following the patterns Reviewer #1: yellow color, Reviewer #2: blue color and Reviewer #3: pink color. Other changes, such as deletions or additions, are highlighted in green color.

Reviewer 2 Report

Comments and Suggestions for Authors

Dear authors,

I have reviewed the manuscript socsci-2658036 entitled Chilean Disaster Response and Alternative Measures for Improvement. In this paper, authors study the gaps in the current system that contribute to its vulnerability and proposes measures for improvement based on a literature review and case studies conducted in Chile. By reading your paper, governments can better establish clear objectives, plans, competencies, guidelines, and coordination within and across sectors. Personally, I am interested in this research direction. However, the paper needs minor improvements before acceptance for publication.

In general:

1.      Start the article with a brief introduction that provides an overview of the importance of DRM and its relevance in the context of Chile. This will help set the context for the rest of the article..

2.      Before discussing the effective DRM measures, provide some background information on the current state of DRM in Chile. This can include information on previous disasters, the existing legal framework, and any initiatives or policies that have been implemented.

3.      To illustrate the effectiveness of DRM measures, include specific examples or case studies from Chile or other countries. This will help readers understand how these measures have been implemented and their impact on disaster management.

4.      Discuss the challenges and limitations faced in implementing DRM measures in Chile. This can include issues related to decision-making power, administrative structure, standardization of early warning systems, and the need for a clear legal framework.

Details:

1.      Figure 1 and Figure 2 are not clear enough, clearer pictures should be provided and a consistent font and size should be used.

  Comments on the Quality of English Language

1.         Page 1 line 16 : " resulted in loss of life " needs to be replaced with " resulted in the loss of life ".

2.         Page 1 line 22 : " and " needs to be replaced with " , ".

3.         Page 3 line 127 : " In order to " needs to be replaced with " To ".

4.         Page 3 line 128 : " in accordance with " needs to be replaced with " following ".

5.         Page 4 line 172 : " requires clear delineation " needs to be replaced with " requires a clear delineation ".

6.         Page 4 line 182 : " prior to " needs to be replaced with " before ".

7.         Page 5 line 219 : " in " needs to be replaced with " of ".

8.         Page 10 line 429 : " a timely manner " needs to be replaced with " on time ".

9.         Page 11 line 480 : " According " needs to be replaced with " According to ".

Author Response

(The authors gave the same response as above.)

Reviewer 3 Report

Comments and Suggestions for Authors

Overall, this paper needs to be improved as it is somewhat vague and unclear. The autor/s have not defined the objective clearly. The methodology of the paper is not clear. The author/s have provided the details of the Sendai Framework, Disaster risk governance measures and the United Nations Office for Disaster Risk Management (DRM). However, it is not clear how they incorporated them into the methodology. Further in methodology, the author/s have proposed several measures based on Amaratunga et al. (2019). This citation is not included in the references. Also, not clear why the author/s selected this citation/article. Even though the authors have given details about the measures pointed out by Amaratunga et al. (2019), they did not mention how they incorporated them into the methodology. There are several typos in the methodology. It is not clear how the autor/s analyzed the paper. 

The author/s has 3 sections after the methodology (4. Results,  5. Talcahuano city – A good practice example for risk management in Chile and 6. Highlights of the new law). I believe these sections are the results of the study. However, it is not yet clear how they connect them. My suggestion is these sub-sections need to come under results and discussion by modifying the title of sub-sections.

Throughout the paper, there are typos, capitalization issues and problems in the structuring of the paragraphs. There are large paragraphs without any citations. Author/s tend to put information in a point form without analyzing them. It is suggested to go through the editing service.

In addition to the comment here, I marked some comments on specific places on the paper.  Those are in yellow. So please go through them carefully and incorporate them. I recommend major revision. 

 Abstract:

The last sentence of the abstract is vague and not clear.

The keywords such as Early warning; Urban Planning; 18 Chilean Law N° 21.364 are included in the abstract.

1.     Introduction:

Line 64- 65: You haven’t given any information to prove this topic sentence of the paragraph.

Line 69-70: You haven’t given any information to prove this topic sentence of the paragraph. No information about the two disasters.

Line 75- 81: This part needs to go to the end of the introduction. It seems you are talking about the structure of the paper. Your objective is not clear yet. Define it.

Line 82: It is not clear why the new law was introduced. Show the research gap.

Line 89: What do you mean by the new system? Spell out what is ONEMI.

2.     Methodology:

Line 97: You mention about 50 documents analyzed. However, it is not clear why you analyzed 50 and what are they. You need to provide a list or at least name the major documents. What is the time frame of the documents? Are these documents related to Chile or international? Not clear yet.

Line 170: Topic sentence has no meaning.

Line 177-178: This sentence has no connection.

3.     Results:

Line 181: The topic sentence looks like the methodology they follow. If so, they need to move it to the methodology section.

Line 189: Mention the year of this plan.

Line 2000-2008: Add the citation here.

Line 237-239: The topic sentence does not represent the paragraph.

Line 272-291: Author/s tends to put the information in the point form without having an analysis. Write them in paragraphs by summarizing the ideas.

Line 305-323: Similar to the above comment. Don’t put the information in the point form.

Line 322-323: This cannot be a paragraph. You need at least 3 sentences to make a paragraph.

4. Legal background, urban planning, and new perspectives

Line 348-352: Need to cite this section.

Line 377-379: This cannot be a paragraph. You need at least 3 sentences to make a paragraph.

Line 381-391: There is no citation in this paragraph. There are many capitalization issues.

Line 393-398: There is no citation in this paragraph.

Line 400: Avoid capitalization.

Line 405-414: Avoid redundancy and make proper paragraphs. All the points have “This refers to a situation with an impact level that can be managed using......”

Line 418: Change the sub-topic to fit into the paragraph.

Line 221: Capitalize the plan.

Line 433: Avoid capitalization.

5. Talcahuano city – A good practice example for risk management in Chile

Line 442-446: Cite please.

Line 458-464: Cite please.

6. Highlights of the new law

Line 468: Modify the sub-topic.

Line 480-497: As mentioned earlier this citation is not in the reference. Also, not clear why the author/s mentioned it here. If you wanna incorporate the idea of the above citation (Amaratunga et al., 2019), please summarize it.

Line 500: It is not clear what is meant by “risk classification” here.

Line 525: Spell out this word.

Line 534: Put a proper sub-topic. This does not match the content of the paragraph.

Line 552-553: Delete this sentence.  

7. Conclusions

Line 555: Do you mean the current DRM system here? 

Comments on the Quality of English Language

I found many typos and capitalization issues throughout the paper. So, it needs to have English editing. 

Author Response

(The authors gave the same response as above.)

Round 2

Reviewer 2 Report

Comments and Suggestions for Authors

Now it could be accepted for publication. 

Comments on the Quality of English Language

Now it is OK. 

Author Response

We thank both reviewers for their time and constructive comments, which improve the quality of our manuscript significantly.

Reviewer 3 Report

Comments and Suggestions for Authors

Please check line 436,  subtitle 3.2. It should be corrected as Decision-making process

Line 480- The full stop mark should come after the parenthesis

Author Response

(The authors gave the same response as above.)
